# Dataset-Learning Duality and Emergent Criticality

**DOI:** 10.3390/e27090989

**Published:** 2025-09-22

**Authors:** Ekaterina Kukleva, Vitaly Vanchurin

**Affiliations:** 1Artificial Neural Computing, Weston, FL 33332, USA; vitaly.vanchurin@gmail.com; 2Duluth Institute for Advanced Study, Duluth, MN 55804, USA

**Keywords:** dataset-learning duality, emergent criticality, scale-invariance

## Abstract

In artificial neural networks, the activation dynamics of non-trainable variables are strongly coupled to the learning dynamics of trainable variables. During the activation pass, the boundary neurons (e.g., input neurons) are mapped to the bulk neurons (e.g., hidden neurons), and during the learning pass, both bulk and boundary neurons are mapped to changes in trainable variables (e.g., weights and biases). For example, in feedforward neural networks, forward propagation is the activation pass and backward propagation is the learning pass. We show that a composition of the two maps establishes a duality map between a subspace of non-trainable boundary variables (e.g., dataset) and a tangent subspace of trainable variables (i.e., learning). In general, the dataset-learning duality is a complex nonlinear map between high-dimensional spaces. We use duality to study the emergence of criticality, or the power-law distribution of fluctuations of the trainable variables, using a toy and large models at learning equilibrium. In particular, we show that criticality can emerge in the learning system even from the dataset in a non-critical state, and that the power-law distribution can be modified by changing either the activation function or the loss function.

## 1. Introduction

Dualities provide a highly useful technique for solving complex problems that have found applications in many branches of science, most notably in physics. For example, well-known dualities include electric–magnetic duality [1], wave–particle duality [2], target–space dualities [3], Kramers–Wannier duality [4,5], etc. More recent, but less well-known examples include a quantum–classical duality [6], dual path integral [7], and pseudo-forest duality [8], to name a few. The key idea of all physical dualities is to establish a mapping (i.e., a duality mapping) between two systems (e.g., physical theories), which can then be used to study properties (e.g., obtaining solutions) of one system by analyzing the other system, or vice versa.

Perhaps the most well-studied example of a physical duality is the so-called bulk-boundary or holographic duality [9,10], such as AdS–CFT [11,12,13,14]. In AdS–CFT, the mapping is established between the bulk, representing a gravitational theory on the anti-de Sitter (AdS) background, and the boundary, representing a conformal field theory (CFT) without gravity.

Mathematical dualities focus on more formal, abstract transformations preserving algebraic or geometric structures, but are also very useful in physics. For instance, dual vector space duality [15], duality of points and lines in projective geometry [16], Hom and Tensor Duality [17], etc.

In this article, we shall consider a learning duality, which we shall refer to as the dataset-learning duality. It is often convenient to view the dataset as representing the states of non-trainable variables on the ‘boundary’, and learning as representing the dynamics (or changes in states) of trainable variables in the ‘bulk’. Thus, the dataset-learning duality may be considered to be an example of a bulk-boundary duality in the context of learning theory. As we shall argue, the duality is a complex nonlinear mapping between very high-dimensional spaces, but near the equilibrium state, the analysis is greatly simplified. This simplification allows us to apply the dataset-learning duality to study the emergence of criticality in learning systems.

Criticality in physical systems refers to the behavior of these systems at or near critical points, where they undergo phase transitions. These critical points are characterized by dramatic changes in physical properties due to the collective behavior of the system’s components. Understanding phase transitions and criticality is essential for explaining many physical phenomena [18] as well as complex biological phenomena such as biological phase transitions [19]. Recent studies have shown that many physical [20,21,22] and biological [23,24] systems can be modeled as learning systems, such as artificial neural networks. Therefore, understanding the criticality and phase transitions in artificial neural networks may also shed light on the emergence of criticality in physical and biological systems.

The phenomenon of self-organized criticality was investigated using biologically inspired discrete network models, which adapt their topology based on local rules without any central control to achieve a balance between stability and flexibility. In papers [25,26,27,28,29,30], networks reach a critical state, where they exhibit power-law distributions of avalanche sizes and durations, indicative of self-organized criticality (SOC), where the criticality is described by a scale-invariant and power-law distribution of fluctuations.

As for the study of artificial neural networks that can be used to solve real applied problems, significant investigation has been carried out in the works [31,32]. There, the eigenvalue spectra of the weight matrices of a trained neural network were empirically examined, and it was found that well-generalizing models exhibit heavy-tailed (power-law) spectral distributions, suggesting that learning guides networks toward a form of self-organized criticality that acts as an implicit regularization. In another paper [33], the researchers carefully analyzed the learning dynamics of two-layer neural networks in the mean-field limit, showing that as the network width increases, the loss landscape becomes increasingly convex and the Hessian spectrum becomes concentrated, implying that wide networks self-organize into a near-critical regime characterized by flat minima and increased sensitivity to disturbances. At the same time, the authors of another paper [34] analytically demonstrated that even in randomly initialized neural networks, nonlinear activation functions can cause significant deviations from the classical spectra of random matrices, suggesting that architectural nonlinearity alone can initiate the emergence of structured, critical behavior in the neural network.

Our own study of criticality concerns the study of fluctuations in trainable variables and the possibility of their adoption by a power-law distribution in learning equilibrium. The first step in this direction was made in Ref. [35], where the criticality, or a power-law distribution of fluctuations, was derived analytically using a macroscopic, or thermodynamic, description developed in [36]. These results were then confirmed numerically [35] for a classification learning task involving handwritten digits [37]. In this paper, we take another step towards developing a theory of emergent criticality by providing a more microscopic description of the phenomena using the dataset-learning duality. We then test our predictions numerically for a classification task.

The paper is organized as follows. In Section 2, a theoretical model of neural networks and basic notations are introduced. Section 3 is devoted to developing a statistical description of a local dataset-learning duality. In Section 4, the distribution of the variables of the tangent space dual to the boundary (dataset) space is analyzed in the multidimensional case. In Section 5, a toy model with two trainable and two non-trainable variables is introduced, and in Section 6, it is solved for power-law fluctuations of the trainable variables. In Section 7, we present numerical results for some specific power-law distributions obtained for specific compositions of activation and loss functions for a toy and a realistic large model. In Section 8, we summarize and discuss the main results of the paper.

## 2. Neural Networks

Consider an artificial neural network defined as a neural septuple $\left(x,\hat{P},p_{\partial },\hat{w},b,f,H\right)$ [36], where:$x\in R^{N}$, is a vector state of *N* neurons,$\hat{P}$, is a projection to subspace of $N_{\partial }=\mathrm{Tr}(\hat{P})$ boundary neurons,$p_{\partial }\left(x_{\partial }\right)$, is a probability distribution which describes the training dataset,$\hat{w}\in R^{N^{2}}$, is a weight matrix which describes connections between neurons,$b\in R^{N}$, is a bias vector which describes bias in inputs of individual neurons,f(y), is an activation map which describes a nonlinear part of the dynamics where $y=\hat{w}x+b$,$H(x,\hat{w},b)$, is a loss function of both trainable and non-trainable variables.

It is assumed that the bias vector b and weight matrix $\hat{w}$ are the only trainable parameters which can be combined into a single trainable vector q via transformation tensors $W_{ij}^{l}$, and $B_{i}^{l}$, (Einstein summation convention over repeated indices is implied here and throughout the manuscript unless stated otherwise.)(1)wij=Wijlql,bi=Bilql.

In standard neural networks, including feedforward [38], convolutional [39], auto-encoder [40], transformers [41], etc., certain trainable variables can be shared, fixed, or set to zero, but all such architectures can be described using appropriate choices of constant tensors $W_{ij}^{l}$ and $B_{i}^{l}$. Then Equation (Equation 1) can be viewed as a linear map from a *K*-dimensional space of trainable variables to an $N^{2}+N$-dimensional space of weights and biases, where *K* can be much smaller than $N^{2}+N$.

The neural septuple $\left(x,\hat{P},p_{\partial },\hat{w},b,f,H\right)$ defines the three relevant types of dynamics and their relevant time-scales:Activation dynamics describes the updating of neurons on the smallest time-scale, which can be set to one, due to their connection with each other and the forward propagation of a signal from the boundary. Bulk neurons are usually subject to such dynamics,(2)x∂(t+1)=(I^−P^)x(t+1)=(I^−P^)f(w^x(t)+b),
where $\hat{I}$ is the identity matrix. However, in the general case of a fully unconstrained neural network, all neurons can undergo activation dynamics.Boundary dynamics describes updates of boundary neurons $x_{\partial }=\hat{P}x$ on the intermediate time-scales, e.g., once per *L* unit time steps, e.g., drawn from probability distribution $p_{\partial }\left(x_{\partial }\right)$ which describes the dataset.Learning dynamics describes changes in trainable variables q on the largest time-scale ML where *M* is the so-called mini-batch size. For example, for the stochastic gradient descent method,(3)$${\dot{q}}_{i}=q_{i}\left(t+ML\right)-q_{i}\left(t\right)=-\gamma \delta_{ij}\frac{d{\langle H\rangle }_{M}}{dq_{j}}$$
where ${\langle \ldots \rangle }_{M}$ is the averaging over mini-batch and γ is the learning rate. Please note that ML is a unit on the scale of q changes; in further analytical reasoning, for simplicity, we set M=1. We emphasize that the method (Equation 3) implies that the space of trainable parameters is flat globally and coordinates $q_{i}$ are orthonormal, i.e., in the case under consideration $g_{ij}=\delta_{ij}$, where $\delta_{ij}$ is the Kronecker delta.

For example, in the case of the feedforward neural network the weight matrix $\hat{w}$ must be nilpotent and if its degree is *L*, i.e., ${\hat{w}}^{L}=0$, then the deep neural network has *L* layers and the state of boundary neurons can be updated once every *L* time steps.

In what follows, we will be interested in the duality mapping from the boundary space to the tangent space of trainable variables. As we shall see, the map is a composition of the activation and learning passes.

## 3. Dataset-Learning Duality

The main objective of this section is to establish a duality mapping between the tangent space of trainable variables and the boundary subspace of non-trainable variables. For starters, we consider a large neural network, defined in Section 2, in a local learning equilibrium, i.e., when the mean value of the trainable parameters remains nearly constant and the evolution is dominated by stochastic fluctuations [36]. In other words, the subject of the study will be a high-dimensional, yet local, problem that allows for significant simplifications. In particular, this allows us to reduce the effective dimensionality *K* of the space of trainable variables to the dimensionality of the space of non-trainable variables *N*, or even to the dimensionality of the boundary subspace of non-trainable variables $N_{\partial }=\mathrm{Tr}(\hat{P})$. However, we will not consider possible symmetries in the dataset that could potentially reduce the dimensionality further.

The learning dynamics, described by Equation (Equation 3), can be viewed as a map from non-trainable degrees of freedom x to changes in trainable degrees of freedom $\dot{q}$, i.e.,(4)(f,x∂(tL−1),…,x∂(t0),x∂|q)→q˙,
where, after the vertical bar, there is a set of parameters of the mapping, which are fixed. Therefore, in Equation (Equation 4), and further vector q is the mean equilibrium value of the trainable variables vector. Similar notations containing a vertical bar cutting off fixed parameters of mapping are used further. Please note that vector f makes sense of bulk neuron values after the final L-th activation step, which will be used to form the prediction of the neural network. The loss function will explicitly depend only on these values of bulk neurons. In Equation (Equation 4) and further notation x∂(ti) denotes the vector of bulk neurons after *i* steps of activation dynamics.

For example, if the loss function *H* of trainable and non-trainable variables is separable, i.e.,(5)$$H=H_{x}\left(x_{\partial },f\right)+H_{q}\left(q\right),$$
then, the evolution of weights and biases is given byb˙j=−γδji∂Hq∂bi−γ∂fj(yj(tL−1))∂yj(tL−1)∂Hx∂fj+…,(6)w˙jk=−γδji∂Hq∂wimδmk−γδklxl(tL−1)∂fj(yj(tL−1))∂yj(tL−1)∂Hx∂fj+….
where there is no summation over *j*, $y_{j}\left(t_{L-1}\right)=w_{ji}x^{i}\left(t_{L-1}\right)+b_{j}$ is the total argument of the activation function $f_{j}$. Please note that for simplicity of notation, f denotes both the argument of the loss function, formed by bulk neurons after L activation steps, and the function itself if the argument is specified in parentheses after it. Therefore, expressions (Equation 6) describe the first step in back-propagation algorithms, ellipses denote all other back-propagation steps.

The activation dynamics, in turn, makes it possible to establish a connection between the initial state of the vector of non-trainable variables (x∂,x∂(t0)) and its state at later time (x∂,x∂(ti)), i.e.,(7)x∂(t1)=f(y(t0)),x∂(t2)=f(y(t1)),…x∂(tL)=f(y(tL−1))=f(x(t0)|q). Please note that during the activation pass, the input neurons $x_{\partial }$ remain fixed and act as a source for the bulk neurons x∂(ti), which change with time $t_{i}$. As a result, the vector of bulk neurons in any activation step x∂(ti) can be expressed through the vector $x(t_{0})$ before the activation starts by recursively applying the activation function. The arguments in parentheses after f specify the functional form of the mapping; its specific form depends on the variables on which it depends explicitly, and they are specified as arguments. This notation is valid in Equation (Equation 7) and in similar cases throughout the paper.

Composition of the activation (Equation 7) and learning (Equation 4) maps is a map from non-trainable degrees of freedom (x∂,x∂(t0)) at time $t_{0}$ to changes in trainable degrees of freedom $\dot{q}$ at time $t_{L}$, i.e.,(8)(x∂,x∂(t0)|q)→q˙. For example, if the learning dynamics is described by stochastic gradient descent, then the map is given by(9)q˙k(x∂,x∂(t0)|q)=−γδkm∂fj(x(t0),q)∂qm∂∂fj+∂∂qmH(x∂,f|q).
which is a map from *N*-dimensional space of non-trainable variables to *K*-dimensional space of fluctuations of trainable variables. Therefore, the probability distribution $p_{\dot{q}}\left(\dot{q}\right)$ can be expressed as(10)pq˙(q˙|q)=∫px∂x∂(x∂,x∂(t0))δ(K)q˙−q˙(x∂,x∂(t0)|q)dNx(t0).
where the different subscripts are used to emphasize that these are different probability distribution functions, e.g., $p_{\dot{q}}\left(\right)$ and px∂x∂() (which is also apparent from the arguments of these functions). The vertical bar is used for conditional distribution, e.g., $p_{\dot{q}}\left(\dot{q}|q\right)$ and for emphasizing the fixed parameterization of functions, e.g., q˙(x∂,x∂(t0)|q). Also, $\delta^{K}$ denotes the K-dimensional Dirac delta function. Please note that we also abuse notation and denote variables, $\dot{q}$, and function, q˙(x∂,x∂(t0)|q), using the same symbols.

If the bulk neurons are initialized to zeros (or other constant values) at $t_{0}$ time moment before starting of activation for every dataset element, then px∂x∂(x∂,x∂(t0))=px∂(x∂)δ(N∂)(x∂(t0)), and by integrating (Equation 10) over x∂(t0) we obtain(11)pq˙(q˙|q)=∫px∂(x∂)δ(K)q˙−q˙(x∂|x∂(t0),q)dN∂x∂.

If $K>N_{\partial }$, then we should be able to perform a local coordinate transformation(12)$$q_{i}^{\prime }=\Lambda_{i}^{j}q_{j}.$$
so that $K-N_{\partial }$ direction become constraints, i.e., ${\dot{q}}_{i}^{\prime }=0$ for $i=N_{\partial }+1,\ldots ,K$. Please note that there is more than one way to do it.

The tangent vector ${\dot{q}}^{\prime }$ can be projected onto $N_{\partial }$-dimensional dynamical subspace with a projection matrix $\hat{R}$ of size $N_{\partial }\times K$,(13)$${\dot{q}}_{i}^{\prime }=R_{i}^{r}\Lambda_{r}^{j}{\dot{q}}_{j}$$
where all components are dynamical. We believe that the region, where this linear transformation of the trainable variables provides fluctuations along new $N_{\partial }$ coordinate axes in Euclidean space, is quite wide. Let us emphasize that although we are considering a problem local to q, non-locality in the boundary space of neurons $x_{\partial }$ in the general case leads us to the need to introduce curvilinear coordinate system $q^{\prime }$ instead of (Equation 12) to satisfy the requirement of fluctuations only along the $N_{\partial }$ coordinate axes according to considering duality.

Writing down expression similar to Equation (Equation 11) in transformed variables and integrating it over $K-N_{\partial }$ constrained directions we obtain(14)pq˙′(q˙′|q′)=∫px∂(x∂)δ(N∂)q˙′−q˙′(x∂|x∂(t0),q′)dN∂x∂.

If the map q˙′(x∂|x∂(t0),q′) is invertible, then it can be considered to be a true duality, and then the probability distributions are related through Jacobian matrix(15)$$p_{{\dot{q}}^{\prime }}\left({\dot{q}}^{\prime }\right)=p_{x_{\partial }}\left(x_{\partial }\left({\dot{q}}^{\prime }\right)\right)\;{\left|\det\left(\frac{\partial {\dot{q}}_{i}^{\prime }}{\partial x_{\partial }^{j}}\right)\right|}^{-1}.$$ We shall refer to this map as the dataset-learning duality. For non-invertible maps q˙′(x∂|x∂(t0),q′) we can write(16)$$p_{{\dot{q}}^{\prime }}\left({\dot{q}}^{\prime }\right)d{\dot{q}}^{\prime }=\sum p_{x_{\partial }}\left(x_{\partial }\right)dx_{\partial },$$
where there is summation over different $x_{\partial }$ that are mapped to the same ${\dot{q}}^{\prime }$. However, even in this more general case, the contribution from a single term in the summation might dominate (e.g., if $p_{x_{\partial }}\left(x_{\partial }\right)$ dominates for some $x_{\partial }$), and then Equation (Equation 15) would still be approximately satisfied.

In summary, to achieve true local dataset-learning duality via the linear transformation (Equation (Equation 12)) of the trainable variables, the range over which this transformation can accurately identify trainable directions in the entire space q must be sufficiently large, and the Jacobian of the transition in Equation (Equation 15) must be invertible.

## 4. Distribution of Fluctuations

To obtain an expression for the Jacobian in Equation (Equation 15), the gradient descent Equation (Equation 9) must be rewritten in the transformed variables (Equation 12) and (Equation 13):(17)q˙i′(x∂|q)=−γRirΛrkδkmΛlmdH(x∂,f|q′)dqm′=−γRirgrl∂fj(x∂|q′)∂ql′∂Hx∂fj+∂Hq∂ql′,
where $g_{rl}=\Lambda_{r}^{k}\Lambda_{lk}={\left(\Lambda \Lambda^{T}\right)}_{rl}$. (Please note that if Λ is an orthogonal matrix, i.e., the transition to the new trainable variables is carried out through a rotation transformation, then the metric in the new variable space would remain unchanged, i.e., $g_{rl}=\delta_{rl}$.) The Jacobian matrix can be expressed as(18)$$\begin{array}{c} \frac{\partial {\dot{q}}_{i}^{\prime }}{\partial x_{\partial }^{k}}=-\gamma R_{i}^{r}g_{rl}\left(\frac{\partial^{2}f_{j}\left(x_{\partial }|q^{\prime }\right)}{\partial x_{\partial }^{k}\partial q_{l}^{\prime }}\frac{\partial }{\partial f_{j}}+\frac{\partial f_{j}\left(x_{\partial }|q^{\prime }\right)}{\partial q_{l}^{\prime }}\frac{\partial^{2}}{\partial x_{\partial }^{k}\partial f_{j}}\right)H_{x}\left(x_{\partial },f\right). \end{array}$$ By substituting it back to (Equation 15), we obtain an expression for the probability distribution of fluctuations of trainable variables(19)$$p_{{\dot{q}}^{\prime }}\left({\dot{q}}^{\prime }\right)=\gamma^{-N}p_{x_{\partial }}\left(x_{\partial }\left({\dot{q}}^{\prime }\right)\right){\left|\det\left(R_{i}^{r}g_{rl}\left(\frac{\partial^{2}f_{j}\left(x_{\partial }|q^{\prime }\right)}{\partial x_{\partial }^{k}\partial q_{l}^{\prime }}\frac{\partial }{\partial f_{j}}+\frac{\partial f_{j}\left(x_{\partial }|q^{\prime }\right)}{\partial q_{l}^{\prime }}\frac{\partial^{2}}{\partial x_{\partial }^{k}\partial f_{j}}\right)H_{x}\left(x_{\partial },f\right)\right)\right|}^{-1}$$
with three factors:$p_{x_{\partial }}\left(x_{\partial }\right)$—distribution of non-trainable input neurons$\frac{\partial H_{x}}{\partial f_{j}}$; $\frac{\partial^{2}H_{x}}{\partial x_{\partial }^{k}\partial f_{j}}$—Jacobian and Hessian of the loss function$\frac{\partial^{2}f_{j}\left(x_{\partial }|q^{\prime }\right)}{\partial x_{\partial }^{k}\partial q_{l}^{\prime }}$; $\frac{\partial f_{j}\left(x_{\partial }|q^{\prime }\right)}{\partial q_{l}^{\prime }}$—dependence of the neural network (result or prediction) f on the dataset/boundary variables $x_{\partial }$ and trainable variables $q^{\prime }$. The first factor depends directly on the boundary dynamics, i.e., training dataset, the second factor depends on the learning dynamics, i.e., the loss function, and the third factor depends on the activation dynamics, i.e., activation function.

We recall that the transformation to primed variables (Equation 12), only $N_{\partial }$ of which fluctuate, can be carried out in different ways, i.e., the transformation matrix Λ is not unique. In fact, we have at our disposal the entire subspace in which the nonzero vector ${\dot{q}}^{\prime }$ lies, to introduce an arbitrary affine coordinate system. We can carry out linear transformations in this subspace, and they will not change the form of Equations (Equation 17)–(Equation 19). At the same time, we have no reason to prefer one coordinate system to another in this subspace for the requirement of power-law distributions along its axes. This freedom can be used to choose the transformed (or primed) trainable variables in which the probability distribution function approximately factorizes, i.e.,(20)$$p_{{\dot{q}}^{\prime }}\left({\dot{q}}^{\prime }\right)\approx \prod_{i=1}^{N_{\partial }}p_{{\dot{q}}_{i}^{\prime }}\left({\dot{q}}_{i}^{\prime }\right).$$ Then the *m*-th statistical moment for some component of the original (or unprimed) variables over some range of scales is given by(21)〈q˙im〉=∫q˙im(q˙′)pq˙(q˙(q˙′))∂q˙∂q˙′dq˙′=∫(Λ−1)ijq˙j′mpq˙′(q˙′)dq˙′≈≈∫(Λ−1)ijq˙j′m∏i=1N∂pq˙i′(q˙i′)dqi′.
where the relationship between primed and unprimed variables (Equation 12) was used.

We shall also assume that each component of fluctuation in the unprimed variables can be determined by only one single (possibly not the same for all) component in the primed variables, i.e.,(22)$${\dot{q}}_{i}\approx \Lambda_{ij}^{-1}{\dot{q}}_{j}^{\prime },$$
where summation over *j* is absent. Then continuing to calculate *m*-th statistical moment for ${\dot{q}}_{i}$ we obtain(23)〈q˙im〉=∫(Λ−1)ijq˙j′mAjq˙j′kjdq˙j′==∫(Λ−1)ijq˙j′mAj(Λ−1)ijq˙j′kj(Λ−1)ijkj+1d(Λ−1)ijq˙j′=∫q˙impq˙i(q˙i)dq˙i,
where the probability distribution of the corresponding primed variables is given by(24)$$p_{{\dot{q}}_{j}^{\prime }}\left({\dot{q}}_{j}^{\prime }\right)=A_{j}{\dot{q}}_{j}^{\prime k_{j}},{\dot{q}}_{j}^{\prime }\in \left[a,b\right].$$ Thus, a power-law distribution of fluctuations for the original trainable variable $q_{i}$ has the same power-law,(25)$$p_{{\dot{q}}_{i}}\left({\dot{q}}_{i}\right)=\frac{A_{j}{\dot{q}}_{i}^{k_{j}}}{{\left(\Lambda^{-1}\right)}_{ij}^{k_{j}+1}},{\dot{q}}_{i}\in \left[{\left(\Lambda^{-1}\right)}_{ij}a,{\left(\Lambda^{-1}\right)}_{ij}b\right],$$
where the change in normalization is associated with a change in the range over which the statistical moment in unprimed variables is calculated. Thus, assuming scale-invariance of the distribution in the transformed variables ${\dot{q}}^{\prime }$, we concluded that, under the condition of distribution $p_{{\dot{q}}^{\prime }}\left({\dot{q}}^{\prime }\right)$ factorization, its power-law form remains unchanged even for the directly trainable weights and biases, as was observed in the previous research [35].

To summarize this section, we have directly demonstrated that the distributions of fluctuations of the trainable variables depend on all three types of dynamics: boundary, activation, and learning. In the following sections, we will analyze the contribution to the distribution $p_{\dot{q}}\left(\dot{q}\right)$ that comes directly from the Jacobian in Equation (Equation 19), based on the toy model. We will also discuss how the power-law distribution $p_{\dot{q}}\left(\dot{q}\right)$ can be achieved by choosing certain compositions of the activation and loss functions. Furthermore, we will experimentally demonstrate that it is precisely the contribution from the Jacobian that is responsible for the emergence of criticality in fluctuations, both in the toy model and in large models, even when the input data distributions are Gaussian.

## 5. Toy Model

In the previous section, we considered a general multidimensional problem and applied the so-called dataset-learning duality to identify trainable variables in which the scale-invariance is expected. In this section, we will consider a simple example of a two-dimensional problem with both continuous and discrete degrees of freedom.

Consider a neural network consisting of only two neurons: input $x^{1}$ and output $x^{2}$ connected with a single trainable weight $w=w_{21}$ and a bias $b=b_{2}$. In addition, we assume that the output neuron can take only two possible values, i.e., its marginal distribution is a sum of two delta functions,(26)$$p_{x^{2}}\left(x^{2}\right)=\frac{1}{2}\delta \left(x^{2}-X^{+}\right)+\frac{1}{2}\delta \left(x^{2}-X^{-}\right).$$ In this case, we can reduce the two-dimensional problem (i.e., two trainable variables $q_{1}=w$ and $q_{2}=b$ and two non-trainable variables $x^{1}$ and $x^{2}$) to two one-dimensional ones, corresponding to two different values, $x^{2}=X^{+}$ and $x^{2}=X^{-}$. Then, for each one-dimensional problem, we can define a single trainable variable $q_{1}^{\prime }$ that is a linear function of $q_{1}$ and $q_{2}$, and along which fluctuations will occur. At the same time, there will be no fluctuations along the orthogonal direction $q_{2}^{\prime }$ according to dataset-learning duality. Then the transformation matrices in Equations (Equation 12) and (Equation 13) are given by(27)$$\begin{array}{ccc} \;\;\;\;\;\Lambda_{\pm } & = & \left(\begin{array}{cc} \cos\theta_{\pm } & \sin\theta_{\pm }\\ -\sin\theta_{\pm } & \cos\theta_{\pm } \end{array}\right), \end{array}$$(28)$$\begin{array}{ccc} R & = & \left(\begin{array}{cc} 1 & 0 \end{array}\right), \end{array}$$
where the rotation angle $\theta_{\pm }$ corresponds to the state of the output neuron $x^{2}=X^{\pm }$.

Let us assume that the loss function *H* depends on the output of the neural network *f* after only a single step of the activation dynamics, but does not depend explicitly on $q^{\prime }$, i.e.,(29)$$H=H_{x}\left(f|x^{2}\right),$$
and then Equation (Equation 17) for the one-dimensional case can be written as(30)$${\dot{q}}_{1}^{\prime }=-\gamma \frac{\partial f(x^{1}|q^{\prime })}{\partial q_{1}^{\prime }}\frac{\partial H}{\partial f}.$$ If we rewrite the loss function as a function of the argument *y* of the activation function f(y), i.e.,(31)$$y\left(x^{1},q^{\prime }\right)=\left(x^{1}\cos\theta_{\pm }+\sin\theta_{\pm }\right)q_{1}^{\prime }+\left(-x^{1}\sin\theta_{\pm }+\cos\theta_{\pm }\right)q_{2}^{\prime }$$

then(32)$${\dot{q}}_{1}^{\prime }=-\gamma \frac{\partial y}{\partial q_{1}^{\prime }}\frac{\partial H(y|x^{2})}{\partial y}=-\gamma \left(x^{1}\cos\theta_{\pm }+\sin\theta_{\pm }\right)\frac{\partial H(y|x^{2})}{\partial y}.$$
and the expression (Equation 18) is given by(33)$$\frac{\partial {\dot{q}}_{1}^{\prime }}{\partial x^{1}}=-\gamma \cos\theta_{\pm }\frac{\partial H(y|x^{2})}{\partial y}.$$

Finally, we arrive at the expression for the probability distribution of fluctuations of the trainable variable in our toy model,(34)$$p_{{\dot{q}}_{1}^{\prime }}\left({\dot{q}}_{1}^{\prime }\right)=\frac{p_{x_{1}}\left(x_{1}\left({\dot{q}}_{1}^{\prime }\right)\right)}{\left|\gamma \cos\theta_{\pm }\right|}{\left|\frac{\partial H(y|x_{2})}{\partial y}\right|}^{-1},$$
which is similar to the general expression (Equation 19).

## 6. Emergent Criticality

In this section, we shall utilize the dataset-learning duality (see Section 3) to investigate the potential emergence of criticality arising mainly from the Jacobian matrix (see Section 4) within the context of the toy model (see Section 5). The idea is to determine conditions under which the criticality might emerge in the toy model and then verify the results numerically for a toy-model classification problem. Specifically, our aim is to identify compositions of activation and loss functions that give us a power-law dependence of the Jacobian leading to a power-law distribution of fluctuations in the trainable variables.

From conservation of probability in terms of ${\dot{q}}^{\prime }$ (here and below we omit the subscript, so there is only one direction of fluctuations) and *y*, we obtain(35)$$p_{{\dot{q}}^{\prime }}\left({\dot{q}}^{\prime }\right)=p_{y}\left(y\left({\dot{q}}^{\prime }\right)\right)\left|\frac{\partial y}{\partial {\dot{q}}^{\prime }}\right|,$$
where the function $y({\dot{q}}^{\prime })$ is assumed to be invertible. On one hand, the power-law dependence of the Jacobian, i.e.,(36)$$\left|\frac{dy}{d{\dot{q}}^{\prime }}\right|=\frac{1}{A|{\dot{q}}^{\prime }{|}^{k}},A=A\left(w,b\right)>0,k\geq 0,$$
implies two possible differential equations(37)$$\begin{array}{c} {\dot{q}}^{\prime }=\left\{\begin{array}{cc} \mathrm{sign}\left({\dot{q}}^{\prime }\right)\exp\left(Ay+B\right)\;\;\;\; & \mathrm{for}\;k=1\\ \mathrm{sign}\left({\dot{q}}^{\prime }\right){\left(Ay+B\right)}^{\frac{1}{1-k}}\;\;\;\; & \mathrm{for}\;k\neq 1 \end{array}\right. \end{array}$$
for some new A=A(w,b), B=B(w,b), and where the form of expressions for fluctuations depends on their sign. On the other hand, the gradient descent Equation (Equation 32) expressed through *y* implies another differential equation(38)$${\dot{q}}^{\prime }=-\left(Dy-C\right)\frac{\partial H}{\partial q^{\prime }},$$
where(39)$$\begin{array}{cc} C & =\frac{\gamma \cos(\theta_{\pm })}{w}\left(b-w\tan\left(\theta_{\pm }\right)\right),\\ D & =\frac{\gamma \cos(\theta_{\pm })}{w}. \end{array}$$
and for the gradient descent(40)Dy−C>0. In this section, we shall study different compositions of loss and activation functions, i.e., H(f(y)), for which Equations (Equation 37) and (Equation 38) are satisfied and thus the emergence of criticality is expected.

For k=1 Equations (Equation 37) and (Equation 38) can be combined together as(41)$$\frac{dH(f(y))}{dy}=-\mathrm{sign}\left({\dot{q}}^{\prime }\right)\frac{\exp(Ay+B)}{Dy-C},$$ By changing variable z=Ay−AC/D in (Equation 41) and integrating in some region with respect to *z* we obtain(42)$$H\left(f\left(z\right)\right)=-\mathrm{sign}\left({\dot{q}}^{\prime }\right)\frac{1}{D}\exp\left(\frac{AC}{D}+B\right)\int_{z_{0}}^{z}dz^{\prime }\frac{\exp(z^{\prime })}{z^{\prime }},$$
or(43)$$H\left(f\left(y\right)\right)=-\mathrm{sign}\left({\dot{q}}^{\prime }\right)\frac{1}{D}\exp\left(\frac{AC}{D}+B\right)\int_{z_{0}}^{Ay-AC/D}dz^{\prime }\frac{\exp(z^{\prime })}{z^{\prime }},$$ One can show that for |z|≫1 the integral $\int dz^{\prime }\frac{\exp(z^{\prime })}{z^{\prime }}$ can be approximated by the integrand, i.e.,(44)$$H\left(f\left(y\right)\right)\approx -\mathrm{sign}\left({\dot{q}}^{\prime }\right)\frac{\exp(Ay+B)}{A(Dy-C)}.$$

To obtain this result, let us consider the following integral:(45)$$I\left(x\right)=\int^{x}\frac{\exp(z)}{z}dz.$$ In the limit of x≫1, we can integrate by parts (iteratively) to obtain(46)$$I\left(x\right)=\frac{\exp(x)}{x}+\frac{\exp(x)}{x}\sum_{n=1}^{\infty }\frac{n!}{x^{n}}\approx \frac{\exp(x)}{x}+\frac{\exp(x+1)}{x^{2}}\approx \frac{\exp(x)}{x},$$
where we took into account that $\sum_{n=1}^{\infty }\frac{n!}{x^{n}}\approx \frac{e}{x\left(1-\frac{1}{x}\right)}$.

In the opposite limit y≡−x≫1, we obtain(47)$$I\left(y\right)=-\frac{\exp(-y)}{y}+\frac{\exp(-y)}{y}\sum_{n=1}^{\infty }\frac{{\left(-1\right)}^{n+1}n!}{y^{n}}\approx -\frac{\exp(-y)}{y}+\frac{\exp(-y-1)}{y^{2}}\approx -\frac{\exp(-y)}{y},$$
where we took into account that $\sum_{n=1}^{\infty }\frac{{\left(-1\right)}^{n+1}n!}{y^{n}}\approx \frac{e^{-1}}{y\left(1+\frac{1}{y}\right)}$.

By combining (Equation 46) and (Equation 47) we obtain(48)$$I\left(x\right)\approx \frac{\exp(x)}{x}.$$
for |x|≫1.

An arbitrary constant coming from the lower limit of the integral must also be added to the expression. We omit it here and in the following cases for simplicity.

Please note that in the limit |Dy|≪|C|, |A|≫|D/C| and B=log|AC| we obtain an exponential function(49)$$\begin{array}{c} H(f(y))=\exp(Ay) \end{array}.$$

For k∈[0,1) Equations (Equation 37) and (Equation 38) can be combined together as(50)dH(f(y))dy=−sign(q˙′)(Ay+B)αDy−C=−sign(q˙′)AD(Ay+B)α(Ay+B)−B+ACD,
where α≡1/(1−k)>1. If $\left|Ay+B\right|\gg \left|B+\frac{AC}{D}\right|$, then(51)$$\frac{dH(f(y))}{dy}=-\mathrm{sign}\left({\dot{q}}^{\prime }\right)\frac{A}{D}\frac{{\left(Ay+B\right)}^{\alpha }}{Ay+B}=-\mathrm{sign}\left({\dot{q}}^{\prime }\right)\frac{A}{D}{\left(Ay+B\right)}^{\alpha -1},$$
whose solution is given by(52)$$H\left(f\left(y\right)\right)=-\mathrm{sign}\left({\dot{q}}^{\prime }\right)\frac{A^{\alpha }}{\alpha D}{\left(y+\frac{B}{A}\right)}^{\alpha }.$$ Please note that for $\left|\frac{A^{\alpha }}{\alpha D}\right|=1$, $\Delta =\frac{B}{A}$ we obtain the following power-law dependence(53)$$\begin{array}{c} H\left(f\left(y\right)\right)={\left(y+\Delta \right)}^{\alpha } \end{array}.$$ In the opposite limit, i.e., $\left|Ay+B\right|\ll \left|B+\frac{AC}{D}\right|$, we obtain(54)$$\frac{dH(f(y))}{dy}=\mathrm{sign}\left({\dot{q}}^{\prime }\right)\frac{A}{D}\frac{{\left(Ay+B\right)}^{\alpha }}{B+\frac{AC}{D}},$$
or(55)$$H\left(f\left(y\right)\right)=\mathrm{sign}\left({\dot{q}}^{\prime }\right)\frac{A^{\alpha +1}{\left(y+\frac{B}{A}\right)}^{\alpha +1}}{\left(\alpha +1)(AC+BD\right)}.$$ For $\left|\frac{A^{\alpha +1}}{\left(\alpha +1)(AC+BD\right)}\right|=1$ and $\Delta =\frac{B}{A}$ the desired composition function takes the form(56)$$\begin{array}{c} H\left(f\left(y\right)\right)={\left(y+\Delta \right)}^{\alpha +1} \end{array}.$$

For k=2 or α=1/(1−k)=−1 Equations (Equation 37) and (Equation 38) can be combined together as(57)$$\frac{dH(f(y))}{dy}=\frac{\mathrm{sign}({\dot{q}}^{\prime })}{\left(Ay+B)(C-Dy\right)}=\frac{\mathrm{sign}({\dot{q}}^{\prime })}{-ADy^{2}+\left(AC-BD\right)y+BC},$$ If the quadratic term in the denominator is small, i.e., |ADy|≪|AC−BD|, then upon integration we obtain(58)$$H\left(f\left(y\right)\right)=\mathrm{sign}\left({\dot{q}}^{\prime }\right)\frac{\log|(AC-BD)y+BC|}{AC-BD}.$$ For |AC−BD|=1, Δ=BC we obtain the following form of composition function(59)$$\begin{array}{c} H(f(y))=-\log|\Delta -y| \end{array}.$$ As can be seen from the formulas given in the boxes, exponential, power-law and logarithmic composition of the activation and loss functions are possible, and all of them, when certain above-mentioned conditions are met, can lead to a power-law contribution from the Jacobian (see Equation (Equation 35)) to the distribution of fluctuations of the trainable variable $q^{\prime }$ under consideration.

## 7. Numerical Results

### 7.1. Toy Model

In this section, we shall present numerical results for the toy model described in Section 5, and compare them with analytical results obtained in Section 6. In particular, we are interested in studying the emergence of critical behavior, or when the distribution of changes in trainable rotated variable $q^{\prime }$ is described by a power-law, i.e.,(60)$$p\left({\dot{q}}^{\prime }\right)\propto {\left({\dot{q}}^{\prime }\right)}^{-k}.$$ The goal is to provide specific numerical examples of the critical behavior (Equation 35) on some ranges of scales for exponential (Equation 49), power-law (Equation 56), and logarithmic (Equation 59) compositions of activation and loss functions, i.e., for H(f(y)).

For the numerical experiments, we consider a simple dataset with only two classes (called ‘−’ and ‘+’) with a single output neuron which takes discrete values ($x^{2}=X^{-}=0$ or $x^{2}=X^{+}=1$). The input subspace also consists of only a single neuron that takes continuous values drawn from two Gaussian probability distributions $p_{-}\left(x^{1}\right)$ (for $x^{2}=0$) and $p_{+}\left(x^{1}\right)$ (for $x^{2}=1$) with mean(61)〈x1〉−=0〈x1〉+=1
where notations ${\langle x^{1}\rangle }_{-}$ and ${\langle x^{1}\rangle }_{+}$ mean averaging over the distributions of the values of the input neuron $x^{1}$ with conditions $x^{2}=X^{-}$ and $x^{2}=X^{+}$, respectively. The standard deviations are chosen as follows(62)〈(x1)2〉−−〈x1〉−2=〈(x1)2〉−=0.25〈(x1)2〉+−〈(x1)〉+2=〈(x1)2〉+−1=0.25. This simple dataset allows us to use the architecture of the toy model of Section 5 with one input neuron $x^{1}$, one output neuron $x^{2}$, one trainable bias *b*, and one trainable weight *w* (from input neuron to output neuron).

Please note that the main goal of this analysis is not to obtain high prediction accuracy (although it is above 97% in all experiments), or to develop an architecture for a realistic classification problem (although a similar dataset would have been obtained if we considered a truncated MNIST dataset [37] with only images of ‘zeros’ and ‘ones’ and input states projected down to a single dimension). Rather, our aim is to demonstrate the emergence of criticality within a simple low-dimensional problem, but we expect the same mechanisms to be responsible for the emergence of criticality in higher-dimensional problems [35].

Also, note that in the learning equilibrium, the average value of the trainable parameters does not change significantly. Therefore, the distribution of the actual changes in trainable variables, i.e., $p(\dot{q})$, and the distribution of potential jumps when the trainable parameters remain fixed, i.e., $p\left(-\gamma \frac{\partial H}{\partial q}\right)$, are essentially the same. However, the latter is more suitable, i.e., we will consider potential jumps caused separately by ‘−’ and ‘+’ classes, which, in the learning equilibrium, can be described as a two one-dimensional problem.

In the remainder of the section, we shall consider five experiments, two for exponential (Equation 49), two for power-law (Equation 56), and one for logarithmic (Equation 59) composition of activation and loss functions.

#### 7.1.1. Exponential Composition, k=1

Consider the sigmoid activation function(63)$$f={\left(1+e^{-y}\right)}^{-1}$$
with mean-squared loss(64)$$H\left(f,x^{2}\right)={\left(f-x^{2}\right)}^{2}$$
or cross-entropy loss(65)$$H\left(f,x^{2}\right)=-\left(1-x^{2}\right)\log\left(1-f\right)-x^{2}\log\left(f\right).$$

For composition of sigmoid activation (Equation 63) and mean-squared loss (Equation 64) functions and ‘−’ class, in the limit y<0 Equation (Equation 63) reduces to $f\approx e^{y}$ and then the composition function is(66)$$H\left(f\left(y\right)\right)\approx e^{2y}.$$ Likewise, for the same composition and ‘+’ class, in the limit y>0 Equation (Equation 63) reduces to $f\approx 1-e^{-y}$ and then the composition function is(67)$$H\left(f\left(y\right)\right)\approx e^{-2y}.$$

Below are the graphs showing the experimental dependencies used to compute the coefficients A,B,C,D for both classes in this case. Knowing these coefficients, one can be confident that the conditions required to obtain the exponential composition (Equation 49) are satisfied. Moreover, it is clear from the graphs in Figure 1 that the mapping between *y* and ${\dot{q}}^{\prime }$ is invertible for chosen points, i.e., duality between these variables is observed. At the same time, from graphs in Figure 2 it is clear that the linear transformation of variables $\left(\dot{q}\right)\to {\dot{q}}^{\prime }$ provides a good approximation of the curve along which fluctuations actually occur. We also carried out a similar procedure to verify the relationships between the coefficients A,B,C,D in subsequent experiments, selecting for analysis only those regions where true dataset-learning duality is realized.

In Figure 3, we plot in logarithmic axes the distributions of fluctuations of $q^{\prime }$ (blue dots) and the contribution from the Jacobian (red dots) to these distributions corresponding to Equation (Equation 35). The linear behavior with slope −1 (or k=1) is in good agreement with Equation (Equation 49).

For composition of sigmoid activation (Equation 63) and cross-entropy loss (Equation 65) function we obtain, for ‘−’ class and in the limit y<0,(68)$$H\left(f\left(y\right)\right)=-\log\left(1-f\left(y\right)\right)\approx e^{y},$$
and for ‘+’ class and in the limit y>0,(69)$$H\left(f\left(y\right)\right)=\log\left(f\left(y\right)\right)\approx e^{-y}.$$ In Figure 4, we plot the distribution of fluctuations of $q^{\prime }$, which is once again in agreement with Equation (Equation 49). Please note that in this case, despite a nearly perfect power-law contribution from the Jacobian (red dots), the distribution of fluctuations as a whole (blue dots) is significantly distorted by the contribution from $p_{y}\left(y\left({\dot{q}}^{\prime }\right)\right)$ in Equation (Equation 35).

We conclude that for the toy-model classification problem with a sigmoid activation function and either mean-squared error or cross-entropy loss functions, the fluctuations of trainable variables can follow a power law with k=1, as confirmed both numerically and analytically.

#### 7.1.2. Power-Law Composition, $k=0;\frac{2}{3}$

Consider a composition of a ReLU activation function(70)f(y)=max(0,y)
and the power-law loss function(71)$$H\left(f,x^{2}\right)={\left(f-x^{2}\right)}^{n}.$$ In this case, non-vanishing fluctuations can take place only when the argument of ReLU is positive, i.e.,(72)y>0,
then, the composition function is also a power law(73)$$H\left(f\left(y\right)\right)={\left(y-x^{2}\right)}^{n}.$$

For n=2, i.e., the standard mean-squared loss, Equation (Equation 56) implies that(74)$$k=\frac{\alpha -1}{\alpha }=\frac{n-2}{n-1}=0,$$
which is in agreement with the numerical results plotted in Figure 5.

For n=4, Equation (Equation 56) implies that $k=\frac{2}{3}$ in agreement with the numerical results plotted in Figure 6.

Thus, we have demonstrated that the power-law composition case (Equation 56) is achieved by combining power-law loss and ReLU activation functions. More generally, we can obtain any $k=\frac{n-2}{n-1}$, where *n* is the power that appears in (Equation 73).

#### 7.1.3. Logarithmic Composition, k=2

Consider the cross-entropy loss function (Equation 65) and a piece-wise linear activation function(75)$$f\left(y\right)=\left\{\begin{array}{c} y,0<y<1,\\ 0,y<0\;\mathrm{or}\;y>1. \end{array}\right.$$ Within the range 0<y<1, the composition of activation and loss function is(76)$$H\left(f\left(y\right)\right)=-\left(1-x_{2}\right)\log\left(1-y\right)-x_{2}\log\left(y\right).$$
which is the logarithm composition of Equation (Equation 59) for ‘−’ class (i.e., $x^{2}=0$) with |Δ|=1, and for ‘+’ class (i.e., $x^{2}=1$) with Δ=0.

In Figure 7, we plot the distribution of fluctuations of $q^{\prime }$, for which k=2, in good agreement with the analytical results in (Equation 59).

### 7.2. Large Model

Next, we describe a procedure that reveals the power-law distribution of fluctuations in the trainable variables of a large neural network, even in the absence of a power-law distribution in the input data.

First, let us address the input data by excluding directions in the data space along which the values of non-trainable variables vary weakly. To achieve this, we compute the covariance matrix $C_{xx}$, perform its spectral decomposition $C_{xx}=SAS^{T}$, and switch to the basis of its eigenvectors, where the covariance matrix becomes diagonal: $C_{x^{\prime }x^{\prime }}=A$.(77)$$\begin{array}{cc} & C_{xx}=E\left[{xx}^{T}\right]=SAS^{T},\\ & x^{\prime }=S^{T}x,\\ & C_{x^{\prime }x^{\prime }}=E\left[{x^{\prime }x^{\prime }}^{T}\right]=E\left[\left(S^{T}x\right){\left(S^{T}x\right)}^{T}\right]=S^{T}C_{xx}S=A. \end{array}$$

Now, we define the transformed vector as $x^{\prime }=S_{cut}^{T}x$, where the matrix $S_{cut}^{T}$ consists of the first *n* rows of $S^{T}$. In other words, the components of $x^{\prime }$ are the projections of the original vector x onto the eigenvectors of the covariance matrix $C_{xx}$ corresponding to its largest *n* eigenvalues. Thus, we work in the subspace of the input data most relevant for learning. In this way, one can effectively estimate the dimensionality of the input data space, which in practical problems is often much smaller than the total number of input neurons.

Next, we calculate the cross-covariance matrix $C_{x^{\prime }\dot{q}}$ of the input data in the transformed variables $x^{\prime }$ with fluctuations of the trainable variables $\dot{q}$. After that, we perform SVD decomposition $C_{x^{\prime }\dot{q}}=U\Sigma V^{T}$ and move to the bases of the left and right singular vectors in the input space and space of trainable variable fluctuations, respectively. In these new variables cross-covariance matrix $C_{x^{\prime \prime }{\dot{q}}^{\prime }}$ has nonzero elements only on the main diagonal.(78)$$\begin{array}{cc} & C_{x^{\prime }\dot{q}}=E\left[{x^{\prime }\dot{q}}^{T}\right]=U\Sigma V^{T},\\ & x^{\prime \prime }=U^{T}x^{\prime },\\ & {\dot{q}}^{\prime }=V^{T}\dot{q},\\ & C_{x^{\prime \prime }{\dot{q}}^{\prime }}=E\left[{x^{\prime \prime }\dot{q}}^{\prime T}\right]=E\left[\left(U^{T}x\right){\left(V^{T}{\dot{q}}^{\prime }\right)}^{T}\right]=U^{T}C_{x^{\prime }\dot{q}}V=\Sigma . \end{array}$$

In such variables, the dataset-learning duality becomes explicit due to the absence of correlations between the input data $x^{\prime \prime }$ and the fluctuations of the trainable parameters ${\dot{q}}_{i}^{\prime }$ for i>dim(x)=n, i.e., now making the transformation $q^{\prime }=V_{cut}^{T}q$, where $V_{cut}^{T}$ consists of the first *n* rows of matrix $V^{T}$, we conclude that the vector $q^{\prime }$ of size *n* and will consider fluctuations along its components.

Note also that the input space $x^{\prime \prime }$ is actually mapped to a manifold that is not flat. However, using the cross-covariance matrix, we can obtain a linear approximation of this generally curved subspace in $\dot{q}$ and gain an understanding of where the input subspace is locally mapped.

Next, we will consider what distribution we have along the direction in input space and what distribution is obtained along the direction in the trainable variables fluctuations space correlated with the chosen direction in $x^{\prime \prime }$. For this, we will select the components ${\left(x^{\prime \prime }\right)}^{1}$ and ${\left({\dot{q}}^{\prime }\right)}_{1}$ along the directions of left and right singular vectors of the cross-covariance matrix in respective spaces, which correspond to the largest singular value of $C_{x^{\prime }\dot{q}}$.

#### 7.2.1. UNN Model

Now let us consider a large fully unconstrained neural network (See Section 2), consisting of 100 input neurons $x_{inp}$ (hereafter, we omit the subscript and denote them simply as x) and 10 output neurons $x_{out}$. The network is fully connected, meaning that every neuron is connected to every other neuron. Please note that there are no bulk (hidden) neurons; instead, both the input and output neurons are updated during the activation dynamics, as they are mutually connected and do not remain fixed. After all steps of activation dynamics (in our example, we used five steps), the values of the output neurons $x_{out}$ are compared with the correct target values for training. The input data x consists of the 100 pixels with the greatest variance across the entire dataset. Thus, the neural network with this architecture is trained to classify images of handwritten digits. Next, the distributions of fluctuations in the local learning equilibrium are analyzed according to the procedure described above.

It can be seen in Figure 8 and Figure 9 that the distribution along ${\left(x^{\prime \prime }\right)}^{1}$ has a form close to a Gaussian distribution, while the distribution of fluctuations along ${\left({\dot{q}}^{\prime }\right)}_{1}$ has an explicit power form in a wide range of scales. In Figure 8, the hyperbolic tangent was used as the activation function for all neurons, and the cross-entropy form of the loss function was chosen, while in Figure 9, the ReLU activation function was used everywhere, and the mean-squared loss function was applied.

#### 7.2.2. CNN Model

Now, let us consider another example of a convolutional neural network, which consists of 2 convolutional layers and two fully connected layers. The total number of trainable variables is 10,000. This neural network, as in other examples, is used to classify the MNIST dataset. The values of all available pixels for each image of a digit, $N_{\partial }=28^{2}$, are fed to the boundary neurons as input data.

Distributions of input data and fluctuations of the trainable variables in the local learning equilibrium for the CNN are shown in Figure 10 and Figure 11. The results are similar to those obtained for the UNN.

It is worth noting that we have already obtained such distribution powers k≈1 and k≈0 analytically and experimentally for the toy model. Interestingly, the experimental data suggest that the presence of criticality and the value of the distribution power do not depend on the size of the neural network. When using the same loss and activation functions in both small and large neural networks, we observed similar powers for the fluctuation distributions at learning equilibrium, namely k≈1 (Figure 4, Figure 8 and Figure 10) and k≈0 (Figure 5, Figure 9 and Figure 11). In all considered models, the power-law distributions of trainable variable jumps do not originate from the input data distributions; rather, their emergence is driven by specific forms of the activation and loss functions.

## 8. Discussion

In this paper, we accomplished two main tasks.

Firstly, we established a duality mapping between the space of boundary neurons (i.e., the dataset) and the tangent space of trainable variables (i.e., the learning), the so-called dataset-learning duality. Both spaces (the boundary and the tangent) are generally very high-dimensional, making the analysis of the duality very non-trivial. However, by considering the problem in a local learning equilibrium and under the assumption that the probability distribution function of the tangent space variables is factorizable, the multidimensional problem can be greatly simplified.

Secondly, we applied the dataset-learning duality to study the emergence of criticality in the learning systems. We show that the observed scale-invariance of fluctuations of the trainable variables (e.g., weights and biases) is caused by the emergence of criticality in the dual tangent space of trainable variables. In particular, we analyzed different compositions of activation and loss function, which can give rise to a power-law distribution of fluctuations of trainable variables on a wide range of scales for a toy model. We showed that the power-law, exponential, and logarithmic compositions of activation and loss functions can all give rise to criticality in the learning dynamics, even if the dataset is in a non-critical state. Main results of the study of criticality are summarized in the following table.
kH(f(y))**Examples**1exp(Ay)Cross-entropy or mean-squared loss functions and sigmoid activation function$\frac{\alpha -1}{\alpha }$${\left(y+\Delta \right)}^{\alpha +1}$Power-law loss and ReLU activation functions2−log|Δ−y|Cross-entropy loss and piece-wise linear activation functions

In addition, we conducted an experiment with a large model classifying digit images from the MNIST dataset. We found that by using the same activation and loss functions as in the toy model, the large neural network exhibits the same power-law behaviors. Thus, we have empirically demonstrated that the results obtained from the toy model generalize to real neural networks. The emergence of criticality and the specific exponents of the trainable variables’ fluctuation distributions depend not on the network size, but on the choice of certain loss and activation functions.

Besides its theoretical significance and potential relevance in modeling critical phenomena in physical and biological systems [18,19], the emergence of criticality is expected to play a central role in machine learning applications. Power-law distributions, in particular, enable trainable variables to explore a broader range of scales without facing exponential suppression. Consequently, criticality is presumed to prevent neural networks from becoming trapped in local minima, a highly desirable property for any learning system. However, the analysis of the learning efficiency and its relation to criticality must involve considerations of non-equilibrium systems, which are beyond the scope of the current paper. Nevertheless, we expect that the established dataset-learning duality can be developed further to shed light on non-equilibrium problems. We leave these and other related questions for future research.

## Figures and Tables

**Figure 1 entropy-27-00989-f001:**
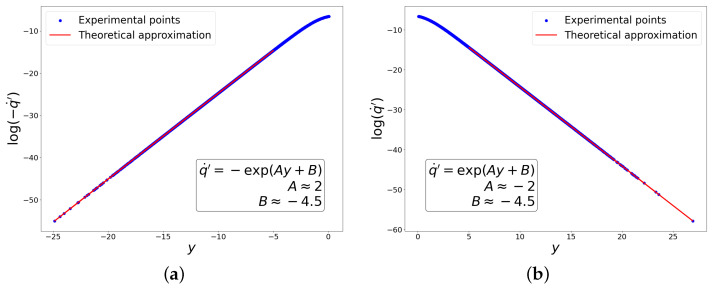
Dependencies for determining coefficients *A* and *B*. (**a**) ‘−’ class. (**b**) ‘+’ class.

**Figure 2 entropy-27-00989-f002:**
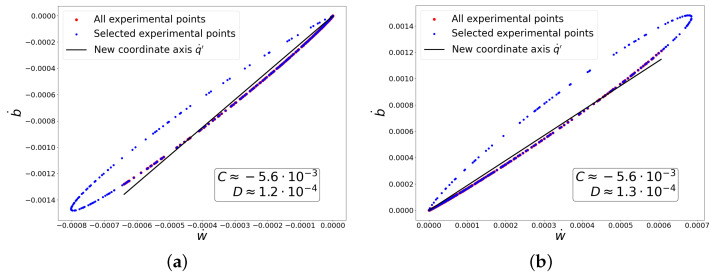
Dependencies for determining coefficients *C* and *D*. (**a**) ‘−’ class. (**b**) ‘+’ class.

**Figure 3 entropy-27-00989-f003:**
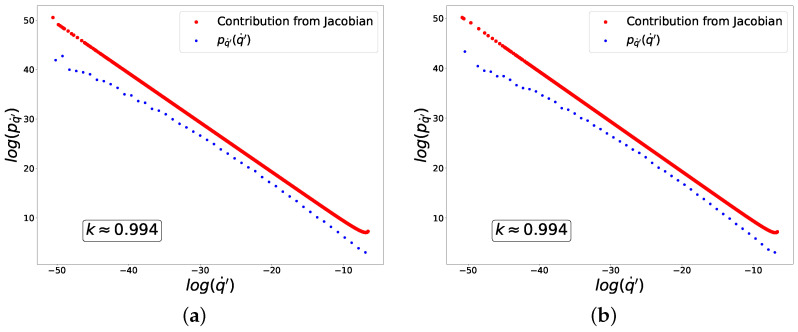
Distribution of fluctuations for composition of sigmoid activation and mean-squared loss functions. (**a**) ‘−’ class. (**b**) ‘+’ class.

**Figure 4 entropy-27-00989-f004:**
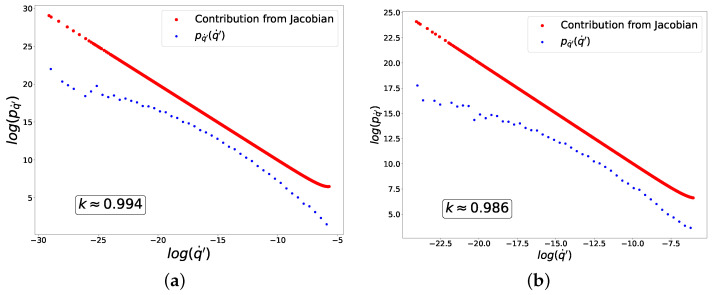
Distribution of fluctuations for composition of sigmoid activation and cross-entropy loss functions. (**a**) ‘−’ class. (**b**) ‘+’ class.

**Figure 5 entropy-27-00989-f005:**
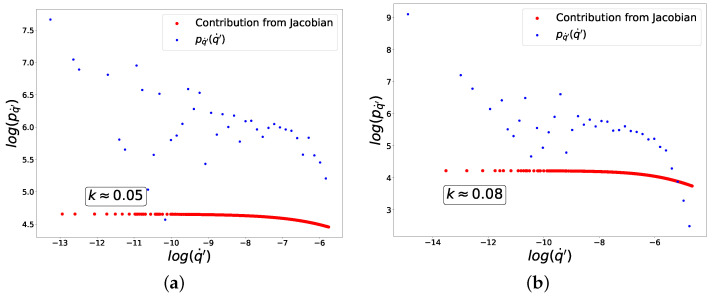
Distribution of fluctuations for composition of ReLU activation and mean-squared loss functions. (**a**) ‘−’ class. (**b**) ‘+’ class.

**Figure 6 entropy-27-00989-f006:**
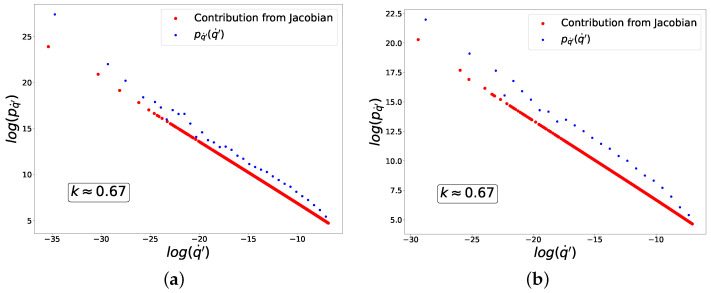
Distribution of fluctuations for composition of ReLU activation and power-law loss function of the fourth power. (**a**) ‘−’ class. (**b**) ‘+’ class.

**Figure 7 entropy-27-00989-f007:**
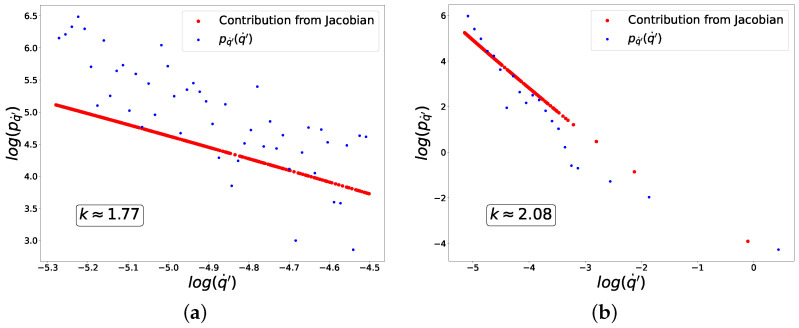
Distribution of fluctuations for composition of piece-wise linear activation and cross-entropy loss functions. (**a**) ‘−’ class. (**b**) ‘+’ class.

**Figure 8 entropy-27-00989-f008:**
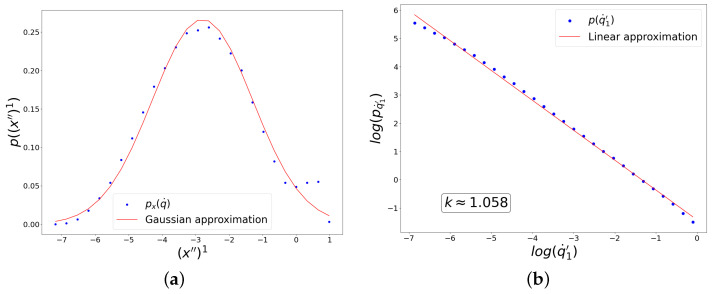
Results of the experiment with the choice of hyperbolic tangent activation and cross-entropy loss functions for UNN. The graphs show distributions of linearly correlated directions in two spaces, which correspond to the largest singular value of the cross-covariance matrix. (**a**) Distribution along the direction in the input data space. (**b**) Distribution of fluctuations of the trainable parameter on logarithmic axes.

**Figure 9 entropy-27-00989-f009:**
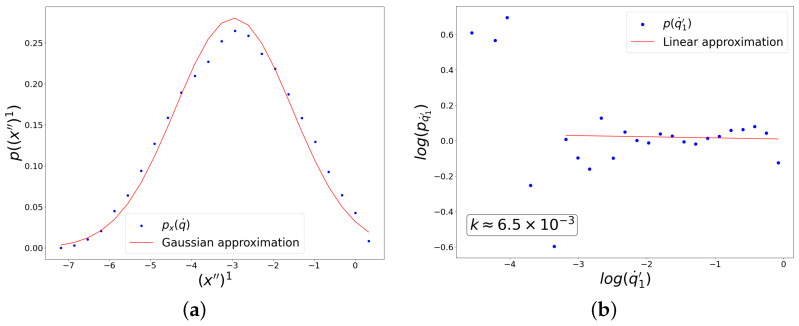
Results of the experiment with the choice of ReLU activation and mean-squared loss functions for UNN. The graphs show distributions of linearly correlated directions in two spaces, which correspond to the largest singular value of the cross-covariance matrix. (**a**) Distribution along the direction in the input data space. (**b**) Distribution of fluctuations of the trainable parameter on logarithmic axes.

**Figure 10 entropy-27-00989-f010:**
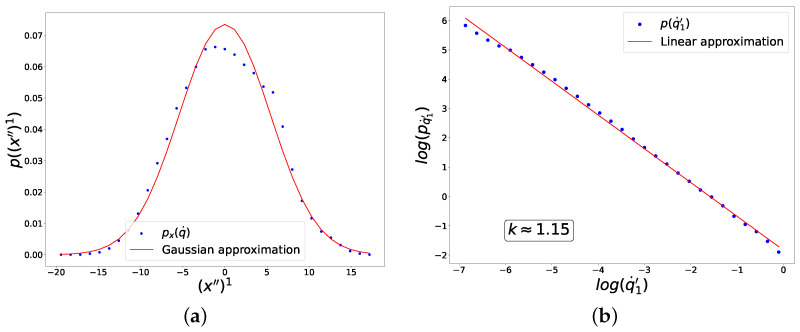
Results of the experiment with sigmoid activation and cross-entropy loss functions for the CNN, showing distributions of linearly correlated directions in two spaces corresponding to the largest singular value of the cross-covariance matrix. (**a**) Distribution along the direction in the input data space. (**b**) Distribution of fluctuations of the trainable parameter.

**Figure 11 entropy-27-00989-f011:**
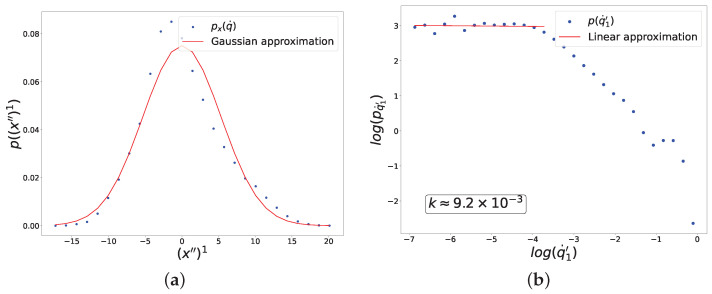
Results of the experiment with ReLU activation and mean-squared loss functions for CNN, showing distributions of linearly correlated directions in two spaces corresponding to the largest singular value of the cross-covariance matrix. (**a**) Distribution along the direction in the input data space. (**b**) Distribution of fluctuations of the trainable parameter on logarithmic axes.

## Data Availability

The data presented in this study are available on request from the corresponding author E.K.

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
