# Peer review of "Dataset-Learning Duality and Emergent Criticality"

_entropy, 2025, doi:10.3390/e27090989_

Round 1
Reviewer 1 Report (Previous Reviewer 3)
Comments and Suggestions for Authors
The manuscript has been improved, but some remarks that were presented for the previous version of the text remain. In particular, the "large model" (section 7.2) remains a "toy" one. The number of neurons is greater, but this is not a shallow nor deep neural network either. At the same time, formulas (77) and (78) do not significantly depend on the number of neurons, but what about network depth and the corresponding attributes based on multiple activations, or other variants of non-linearities in the neural architecture? The research deserves attention, but some important issues are not given enough consideration.
There are a few additional comments:
- The authors should more clearly present the point-by-point contribution of this research to the subject area.
- Scientific novelty should be discussed both in the abstract and the introduction.
Author Response
Please see the attachment.

Reviewer 2 Report (New Reviewer)
Comments and Suggestions for Authors
The presented paper proposes a dataset-learning duality concept and explains how this approach affects the emergence of criticality in neural networks. The main idea is that the activation and learning passes together define a function between the inputs of the networks and the weights and biases. This duality is then used to discuss conditions under which fluctuations of the trainable variables follow power-law distributions.
This work is clearly original and the analogy with bulk-boundary dualities from physics is interesting. However, there are some minor issues that need to be addressed.
Firstly, the authors should define more rigorously what they exactly mean by "local learning equilibrium". It would be very helpful to provide a formal definition. Also, the difference between this equilibrium and the standard convergence of gradient descent method should also be discussed.
Secondly, the duality mapping is said to be true when the determinant of the Jacobi matrix is invertible. But when is the invertibility guaranteed? This is not discussed in detail.
And finally, there are some issues regarding the experiments performed in order to validate this method. The claim that the emergence of criticality is independent of dataset distribution or network size is not sufficiently supported. The analysis should be extended to at least a modern deep learning architecture, such as CNN or Transformer. Also, a short glossary of symbols (from sections 3 and 4 mostly) should be added, as there are heavy notations that not every reader is familiar with.
Author Response
Please see the attachment.

Reviewer 3 Report (New Reviewer)
Comments and Suggestions for Authors
The authors shows that a composition of the two maps establishes a duality map between a subspace of untrainable boundary variables and a subspace of trainable variables. This duality allows to study the emergence of criticality, or the power-law distribution of fluctuations in trainable variables, using toy and large models in learning equilibrium. They shows that criticality can arise in the learning system even from a data set in a non-critical state.
This research outline an very interesting way to show that the learning process is a mechanism that redirects neural connections to critical behavior. This manuscript is suitable for publication in the present version.
Round 2
Reviewer 1 Report (Previous Reviewer 3)
Comments and Suggestions for Authors
All responses are clear.
This manuscript is a resubmission of an earlier submission. The following is a list of the peer review reports and author responses from that submission.
Round 1
Reviewer 1 Report
Comments and Suggestions for Authors
The work is quite interesting from the point of view of an attempt to develop a theory of neural networks.
I think that this work is of no interest for physicists, since the entropy of the system is considered only as a loss function, which is used in the theory of neural networks.
This work is devoted to the process of deep learning of an artificial neural network. Accordingly, it is interesting for researchers (like me) who study artificial neural networks. To train a neural network, it is necessary to choose some loss function - the authors chose entropy for this purpose. This approach is not new, it was used by Geoffrey Hinton in a one-step approximation (backpropagation method for training multilayer neural networks, restricted Boltzmann machine). However, at present, most researchers use simpler and more effective methods for training neural networks.
1. The work is not very relevant but interesting.
​​​​​​​2. I don't know how to use the results of this work. Maybe someone will make an application based on this?
3. The paper well written, the text clear and easy to read.
4. The conclusions consistent with the evidence and arguments presented.
My recommendation is "Weak Accept".
Reviewer 2 Report
Comments and Suggestions for Authors
The manuscript is well written., however there are some points which should be improved. Here is my suggestion:
- While the idea of duality is well-known in physics, its adaptation to machine learning could benefit from a clearer, more intuitive explanation, especially for readers who aren't deeply familiar with theoretical physics. The introduction, though informative, might be too technical for a broader audience to fully grasp.
- The duality framework introduced in the paper is mainly developed under equilibrium conditions and explored through a simplified toy model. However, it’s not clear how well this approach would carry over to more realistic, large-scale, or non-equilibrium learning systems. A discussion on its potential limitations or scalability would help clarify this.
- The suggestion that criticality can arise from non-critical datasets is certainly thought-provoking, but it would be stronger with more rigorous support. The current argument leans heavily on abstract, analytical reasoning—whereas real-world datasets often have complex, high-dimensional structures that may behave differently.
- At times, the mathematical sections become dense very quickly—for instance, in Equation (19)—which might make it difficult for readers to follow the logical flow. The derivations are spread out and involve multiple variable changes, so it would be helpful to streamline these or summarize the key steps more clearly.
- While the references are generally solid, the paper could be strengthened by including more recent machine learning research related to criticality, such as work on spectral properties or heavy-tailed distributions in neural networks.
- Finally, some equations, like Equation (6), are quite packed with symbols. This can overwhelm readers unless each variable is clearly defined. Including inline explanations or a dedicated glossary for terms like f, y, and their derivatives would make the paper much more accessible.
Reviewer 3 Report
Comments and Suggestions for Authors
The paper is devoted to investigating the duality between the spaces of data and the trainable variables in artificial neural networks. A theoretical framework is proposed, and experimental evidence supporting the appearance of criticality is provided. The findings offer important insights into the functioning and training principles of neural networks. It corresponds to the aims and scope of the “Entropy” journal.
However, there are several significant questions regarding scientific contributions. For example, function under determinant in formula (19) has not been studied and its mathematical properties are not discussed. In Section 7, author only suggests the same mechanisms to be responsible for the emergence of criticality in higher-dimensional problems (see lines 316-322). However, no further confirmation is given in the text. An extremely simple model of two neurons is considered as an example while modern AI models have hundreds of billions neurons and studying them with this approach is impossible both analytically and computationally. Author needs to improve results significantly to make research relevant to AI field. At least explanation of algorithm for real neural network and some new asymptotic results are required.
There are a few additional comments:
1. It is unacceptable to merely mention a set of papers (e.g., [8-11], [17-22], etc.). What were the results of the previous studies? Each one should be discussed. Moreover, the authors needs to explain how their new findings fill a research gap relative to known results.
2. The authors should more clearly present the point-by-point contribution of this research to the subject area.
3. Scientific novelty should be discussed both in the abstract and the introduction.
4. Appendix A is too short. It should be moved to Section 6.